# Sustained Oropouche virus transmission in Rio de Janeiro's Atlantic Forest: genomic evidence over a two-year period

Ighor Arantes[1], Fernanda de Bruycker-Nogueira[1], Carla de Oliveira[1], Patrícia Carvalho Sequeira[1], Otília Lupi[2], Michele Borges[2], Carolina Lopes Melo[2], Michelle Brendolin[2], Antonio Braga[3], Luciane de Souza Velasque[3], Andréa Cony Cavalcanti[4], Adriana Cardoso Camargo[4], Fábio Burack da Costa[5], Cristiane Gomes de Castro Moreira[6], Vanessa Zaquieu Dias[6], Thiago de Jesus Sousa[7], Felipe Donateli Gatti[7], Gabriela Colombo de Mendonça[7], Joana Zorzal Nodari[7], Rodrigo Ribeiro-Rodrigues[7,8], Edson Delatorre[9], Guilherme Calvet[2], Felipe Gomes Naveca[1,10], Patricia Brasil[2], Ana Maria Bispo de Filippis[1], Gonzalo Bello[1/+], on behalf of the OROV-Rio de Janeiro Outbreak Response Working Group[*]

[1]Fundação Oswaldo Cruz-Fiocruz, Instituto Oswaldo Cruz, Laboratório de Arbovírus e Vírus Hemorrágicos, Rio de Janeiro, RJ, Brasil
[2]Fundação Oswaldo Cruz-Fiocruz, Instituto Nacional de Infectologia Evandro Chagas, Laboratório de Doenças Febris Agudas, Rio de Janeiro, RJ, Brasil
[3]Secretaria de Estado de Saúde do Rio de Janeiro, Rio de Janeiro, RJ, Brasil
[4]Laboratório Central de Saúde Pública do Estado do Rio de Janeiro, Rio de Janeiro, RJ, Brasil
[5]Fundação Oswaldo Cruz-Fiocruz, Instituto Oswaldo Cruz, Laboratório das Interações Vírus-Hospedeiros, Rio de Janeiro, RJ, Brasil
[6]Universidade Federal do Rio de Janeiro, Instituto de Microbiologia Paulo de Góes, Departamento de Virologia, Rio de Janeiro, RJ, Brasil
[7]Laboratório Central de Saúde Pública do Estado do Espírito Santo, Vitória, ES, Brasil
[8]Universidade Federal do Espírito Santo, Núcleo de Doenças Infecciosas, Vitória, ES, Brasil
[9]Universidade Federal do Espírito Santo, Centro de Ciências da Saúde, Departamento de Patologia, Laboratório de Genômica e Ecologia Viral, Vitória, ES, Brasil
[10]Fundação Oswaldo Cruz-Fiocruz, Instituto Leônidas e Maria Deane, Núcleo de Vigilância de Vírus Emergentes, Reemergentes ou Negligenciados, Manaus, AM, Brasil

**BACKGROUND** Oropouche virus (OROV), an arbovirus endemic to the Amazon region, has recently expanded into non-endemic areas including Rio de Janeiro State, Brazil.

**OBJECTIVE** To characterise the spatio-temporal dynamics and ecological factors associated with OROV transmission in Rio de Janeiro during 2024-2025.

**METHODS** We analysed OROV case-associated ecological factors and performed a phylodynamic analysis on 40 viral genomes, comprising 35 new and five published sequences, sampled from 15 municipalities across the state during 2024-2025.

**FINDINGS** OROV cases showed significant positive correlations with forest area (r = 0.50, p < 0.0001), banana harvest area (r = 0.39, p < 0.01), and cassava harvest area (r = 0.29, p < 0.05); but these factors were autocorrelated, suggesting a confounded relationship. We identified two OROV sub-clades circulating in the Rio de Janeiro State. The OROV$_{RJ/ES}$ sub-clade was likely introduced into the Southern Fluminense region around January 2024, spread primarily by short (> 2 km, 50% of events) and mid-distance movements (2-9 km; 30%) with a mean dispersal rate of 0.3 km/day, and seed outbreaks in Metropolitan and Northwest Fluminense regions in 2025. The OROV$_{ES-I}$ clade was likely introduced into Central Fluminense and Coastal Lowlands regions, later spreading to the Northern Fluminense region.

**MAIN CONCLUSIONS** Following its introduction in early 2024, OROV persisted in Rio de Janeiro State by spreading through short-distance movements among municipalities with high forest cover and agricultural areas. The sustained multi-year OROV transmission in the Atlantic Forest biome highlights the potential for establishment of endemic cycles beyond the Amazon region and the need for enhanced surveillance in extra-Amazonian areas, where OROV will evolve in a different ecosystem.

Key words: Oropouche virus - Oropouche fever - Brazil - Rio de Janeiro - Atlantic Forest

**doi:** 10.1590/0074-02760250181
**Financial support:** CNPq [grant CNPq/MCTI 10/2023 - Faixa B - Grupos Consolidados - Universal 2023 (421620/2023-4), PROEP-IOC-CNPq (442176/2024-4)], FAPERJ (E-26/211.565/2019).
IA had support from FAPERJ (grant SEI-260003/019669/2022), GB is supported by CNPq through their productivity research fellowship (304883/2020-4), FGN is supported by CNPq productivity research fellowship (307748/2025-1), PB was supported by CNPq and FAPERJ through their productivity research fellowships (311562/2021-3 and E-26/200.935/2022), ED and RRR are supported by a grant from the State of Espírito Santo Government through the FAPES (grant no. 2025-25N6P-DI 006/2025-SESA/SEAG/FAPES).
IA and FB-N contributed equally to this work.
**+ Corresponding author:** gbello@ioc.fiocruz.br | ⓘ https://orcid.org/0000-0002-2724-2793

**Handling editor:** Mariza Morgado | ⓘ https://orcid.org/0000-0002-9304-6184

*Members of the OROV-Rio de Janeiro Outbreak Response Working Group: Cintia Damasceno dos Santos Rodrigues, Carolina Cardoso dos Santos, Juan Carlos Proença Moura, Desiree dos Santos Nunes, Simone Alves Sampaio (Laboratório de Arbovírus e Vírus Hemorrágicos, Instituto Oswaldo Cruz, FIOCRUZ, Rio de Janeiro, Brazil); Claudia Maria Braga, Mario Sergio Ribeiro, Debora Fontenelle dos Santos, Silvia Cristina de Carvalho Cardoso, Cristina Giordano, Paula Almeida, (Secretaria de Estado de Saúde do Rio de Janeiro); Lusiele Guaraldo, Luana Damasceno, Thais Pires Trindade, Leticia Lopes Corrêa, Fernanda Moronoe, Heloisa Ferreira, Trevon Fuller, Isabella Moraes, Marise Mattos, Ezequias Batista Martins, Rogério Valls, Anielle Pina Costa, Roxana Flores, Clarisse Bressan, Manuela da Costa Medeiros, Stephanie Penetra, Otávio de Melo Espíndola, (Laboratório de Doenças Febris Agudas, Instituto Nacional de Infectologia Evandro Chagas, FIOCRUZ, Rio de Janeiro, Brazil); Ieda Pereira Ribeiro, Myrna Cristina Bonaldo (Laboratório de Medicina Experimental e Saúde - IOC/Fiocruz - RJ). Elizabeth Portari (Instituto Fernandes Figueira, Fiocruz, RJ).

Oropouche virus (OROV) is an arbovirus belonging to the *Peribunyaviridae* family, genus *Orthobunyavirus*, that causes Oropouche fever, a neglected tropical disease that is often clinically indistinguishable from other arboviral diseases.[1] Unlike other urban arboviruses circulating in Latin America, OROV is endemic to the Amazon biome and is primarily transmitted to humans by the biting midge *Culicoides paraensis*, a vector adapted to both sylvatic and semi-urban environments.[2] The OROV genome consists of three single-stranded, negative-sense RNA segments: L (large), M (medium), and S (small), which enable rapid viral evolution via segment reassortment during mixed infections.[3] Between late 2022 and early 2024, four Brazilian states in the western Amazon region (Amazonas, Acre, Rondônia, and Roraima) experienced extensive OROV outbreaks linked to the spread of a newly reassortant viral lineage, designated $OROV_{BR-2015-2025}$.[4]

In 2024, the first confirmed OROV outbreaks outside the Brazilian Amazon were reported in several small municipalities within the Atlantic Forest biome.[5] These outbreaks were caused by the novel reassortant $OROV_{BR-2015-2025}$ clade that spread multiple times from the Amazon basin to non-endemic Atlantic Forest regions of Brazil during early 2024, successfully establishing local transmission in Southern, Southeastern, and Northeastern states.[6,7] Viral sub-clades originating in the State of Amazonas, designated $OROV_{AM-I}$ and $OROV_{AM-II}$, seeded outbreaks in Santa Catarina ($OROV_{SC-I}$) and Espírito Santo ($OROV_{ES-I}$), respectively. Meanwhile, a third sub-clade circulating in the Northern states of Acre and Rondônia, called $OROV_{AMACRO-II}$, gave rise to lineages detected in Santa Catarina ($OROV_{SC-II}$), Espírito Santo ($OROV_{ES-II}$ and $OROV_{ES-III}$), and across both Rio de Janeiro and Espírito Santo ($OROV_{RJ/ES}$).

The Brazilian State of Rio de Janeiro reported a small OROV outbreak in 2024 ($n = 151$), mainly in the Southern Fluminense region, followed by a sharp increase in cases in 2025 ($n = 2,463$), particularly in rural areas of the Metropolitan, Southern, and Northern Fluminense regions.[8] It remains unclear whether the 2025 epidemic in Rio de Janeiro resulted from a new viral introduction or from ongoing transmission of the $OROV_{RJ/ES}$ lineage introduced in 2024. To clarify this, we conducted genomic surveillance with phylodynamic analyses of OROV circulating in Rio de Janeiro State in 2024-2025.

## MATERIALS AND METHODS

*OROV-positive samples and ethics* - As part of a collaborative surveillance effort involving the Rio de Janeiro State Central Laboratory (LACEN/RJ), the Instituto Nacional de Infectologia Evandro Chagas (INI) at FIOCRUZ, Rio de Janeiro, and the Laboratório de Arbovirus e Vírus Hemorrágicos at Instituto Oswaldo Cruz, FIOCRUZ, Rio de Janeiro, 35 OROV-positive samples identified across different municipalities of the Rio de Janeiro State between 2024 and 2025 using a duplex real-time reverse transcription polymerase chain reaction (RT-PCR) assay that detects Oropouche and Mayaro viruses,[9] were selected for near-full-genome sequencing [Supplementary data (Tables I-II)]. This study was approved by the Ethics Committee of Instituto Oswaldo Cruz (CAAE: 90249218.6.1001.5248). Access to the genetic heritage of the OROV under investigation is registered in the National System for the Management of Genetic Heritage and Associated Traditional Knowledge (SisGen A0C0D2F).

*Whole-genome sequencing and genome assembling* - OROV-positive samples were submitted to total RNA extraction using the Thermo Scientific™ KingFisher™ Flex Purification System and used in sequencing library preparation following an adapted version of Illumina's COVIDseq assay previously optimised for OROV sequencing.[4] Library preparation was carried out using Illumina's COVIDSeq kit, and sequencing was performed on a MiniSeq version 2.3.0 instrument with the Mid-Output Reagent Cartridge kit (300 cycles) for a 151 bp × 2 paired-end run. The procedure was conducted at the Genomic Platform - Next Generation Sequencing - RPT01J (Fiocruz Technological Platforms Network). FASTQ reads were generated on Illumina BaseSpace (https://basespace.illumina.com), and consensus genomes were assembled using ViralFlow version 1.2.0.[10] Reference sequences AF484424, AF441119, and AY237111 from GenBank were used for the L, M and S, virus segments of OROV, respectively. Consensus genomes are available at GISAID (https://doi.org/10.55876/gis8.250523gx).

*OROV whole-genome genotyping* - Near-full-length ($\geq 70\%$ of coverage) OROV consensus sequences of the S, M, and L genomic segments from Rio de Janeiro, both generated in this study ($n = 35$) and previously published ($n = 5$), were genotyped following the method described by Naveca et al.[4] All RJ genomes were aligned using MAFFT v7.505[11] against curated reference datasets specific to each designated OROV genome segment. The datasets included (I) a subset of all genomes previously assigned to each segment type (L1/L2, M1/M2, S1/S2/S3)[4] clusterised with CD-HIT v.2.4.8;[12] (II) a clustered subset of all $OROV_{BR-2015-2025}$ ($L_2M_1S_2$) clade genomes available until March 2024; and (III) the following prototype virus sequences, all classified within the species *Orthobunyavirus oropoucheense*: OROV (GenBank accessions L: AF484424, M: AF441119, S: AY237111), Iquitos virus (L: KF697142, M: KF697143, S: KF697144), Perdões virus (L: KP691627, M: KP691628, S: KP691629), and Madre de Dios virus (L: KF697147, M: KF697145, S: KF697146). The datasets were used to infer maximum likelihood (ML) trees for each OROV genomic segment using IQ-TREE v2.2.2.7[13] under the best nucleotide substitution model selected by the ModelFinder software.[14] Branch support was assessed using the approximate likelihood-ratio test (aLRT)[15] based on the Shimodaira-Hasegawa-like procedure with 1,000 replicates. The ML trees were visualised using Figtree v.1.4.4,[16] and the Treeio v3.1.7,[17] and the ggtree v3.2.1[18] R packages. Genotypes were defined by the clustering of RJ sequences with reference sequences exhibiting robust statistical support (aLRT > 0.90).

*$OROV_{BR-2015-2025}$ subclade assignment* - All OROV genomes from Rio de Janeiro State assigned to the $OROV_{BR-2015-2025}$ clade were further resolved into its subclades.[4] To construct a dataset for this analysis, represent-

ative sequences from established subclades (AMACRO$_I$, AMACRO$_{II}$, AM$_I$, AM$_{II}$, AM$_{III}$, and RR$_I$) were selected by a clusterisation approach using CD-HIT v.2.4.8[12] using variable cutoff values to obtain datasets of equivalent size. This dataset was supplemented with the oldest known genome from this lineage, sampled in Tefé, Amazonas, in 2015. A ML phylogenetic tree was then inferred using the parameters previously described. The assignment of sequences from Rio de Janeiro to specific sub-clades was determined by the formation of a monophyletic group with reference sequences from that sub-clade, supported by robust statistical evidence (aLRT > 0.90).

*Bayesian discrete phylogeographic analysis* - To elucidate the introduction pathways and subsequent interstate dissemination dynamics of OROV in Rio de Janeiro State, a comprehensive phylogeographic dataset was assembled. This dataset comprised: (I) all available OROV sequences originating from Rio de Janeiro State; (II) all available sequences from the parental clades of these Rio de Janeiro sequences, located in the Amazonian regions; and (III) the two chronologically earliest genomes identified within the OROV$_{BR-2015-2024}$ subclade, which were sampled in Tefé, Amazonas (2015), and French Guiana (2020) [Supplementary data (Table III)]. The temporal structure of this selected dataset was evaluated using TempEst v1.5.3[19] through a root-to-tip linear regression. The significance of the correlation between collection date and genetic distance was assessed using a Pearson correlation test, with adjustments for multiple comparisons made via the Benjamini-Hochberg method for false discovery rate control. Time-scaled phylogenetic trees were estimated using the Markov Chain Monte Carlo (MCMC) approach implemented in BEAST v1.10.4,[20] with BEAGLE v4.0.0[21] employed to improve computational efficiency. Bayesian trees were reconstructed under the nucleotide substitution model selected by the ModelFinder application,[14] the non-parametric Bayesian Skyline coalescent demographic model,[22] and a relaxed molecular clock model with a continuous-time Markov chain (CTMC) rate reference prior.[23] The time to the most recent common ancestor (T$_{MRCA}$), and the spatial dynamics of the identified RJ clusters were reconstructed using a discrete phylogeographic model[24] implemented in BEAST v1.10.4.[20] For this analysis, we employed a CTMC rate reference prior and a symmetric phylogeographic model. Reflecting the focus on reconstructing the introduction and dissemination dynamics of OROV within Rio de Janeiro State, sequences originating from this state were assigned to their most probable region of infection; in contrast, all other sequences (*i.e.*, those not from Rio de Janeiro State) were coded either by their Brazilian state of sampling or, if international, by their country of origin. Adequate mixing and convergence of the MCMC run were confirmed by ensuring that all continuous parameters achieved an effective sample size (ESS) exceeding 200, as verified using Tracer v1.7.[19] The maximum clade credibility (MCC) tree was summarised with TreeAnnotator v1.10[20] and visualised using Figtree v.1.4.4[16] and also the Treeio v3.1.7,[17] and ggtree v3.2.1[18] R packages. The spatial dis-

semination patterns of major OROV clusters across the administrative regions of Rio de Janeiro State were geographically visualised on maps generated using the ggplot2 (17), sf v.1.0-20,[25] and geobr v.1.9.1[26] R packages.

*Bayesian continuous phylogeographic analysis* - The intrastate OROV dissemination dynamics in Rio de Janeiro and its diffusion rate were additionally inferred using a continuous phylogeographic analysis employing a heterogeneous relaxed random walk model with a Cauchy distribution.[27] For this analysis, an inclusion criterion was established whereby only clusters comprising more than 30 genomes sampled within Rio de Janeiro State were selected. This threshold was implemented to ensure the robust estimation of cluster-specific epidemiological parameters associated with viral dissemination and to more accurately capture the dynamics of viral circulation within the state. Sequences were coded based on the spatial coordinates of their city of origin, and for sequences from the same municipality, a noise factor (0.01 degrees) was added to the sampling coordinates to ensure each sequence had unique geographic coordinates. The weighted diffusion rate and its variation across branches was inferred using the seraphim R package,[28] and migratory events were summarised with the cross-platform software SPREAD v1.0.7.[29] The spatiotemporal diffusion patterns of the virus and its associated epidemiological parameters were, respectively, projected onto maps and graphically plotted. These visualisations and analyses were conducted in R, utilising the packages geobr v.1.9.1,[26] ggplot2 (17), sf v.1.0-20,[25] geosphere v.1.5-18[30] and HDInterval v.0.2.4.[31]

*Statistical analysis* - We obtained municipal-level data for Rio de Janeiro State on land cover (forest, urban, and grassland area) and the harvested area of the 10 most important agricultural products in 2024 from the public databases of the Instituto Brasileiro de Geografia e Estatística (IBGE).[32,33] We used Spearman's correlation test to assess relationships between OROV case counts and ecological factors, as well as the correlations among the ecological factors themselves, across all municipalities reporting cases. We used the Mann-Whitney test to compare case counts of OROV, dengue, and chikungunya in the 10 municipalities with the highest OROV incidence. Tests were considered statistically significant if the p-value was < 0.05. Statistical analyses were performed using the GraphPad Prism v7.0 (Prism Software, USA).

## RESULTS

A total of 2,614 OROV positive cases were detected in the Rio de Janeiro State during 2024 (*n* = 151) and 2025 (*n* = 2,463) (Fig. 1A). The majority of OROV cases (93%) were concentrated in 26 municipalities experiencing medium (15 - 50 cases) or large (> 50 cases) outbreaks, whereas the remaining 66 municipalities reported few (< 15) or no cases. The highest incidence rates of OROV (> 150 cases/100,000 inhabitants) were concentrated in small, inland municipalities characterised by small populations (< 60,000 inhabitants) and limited urbanised areas (< 30 km²). Notably, in 2025, the mean number of OROV cases (*n* = 106) in those localities was

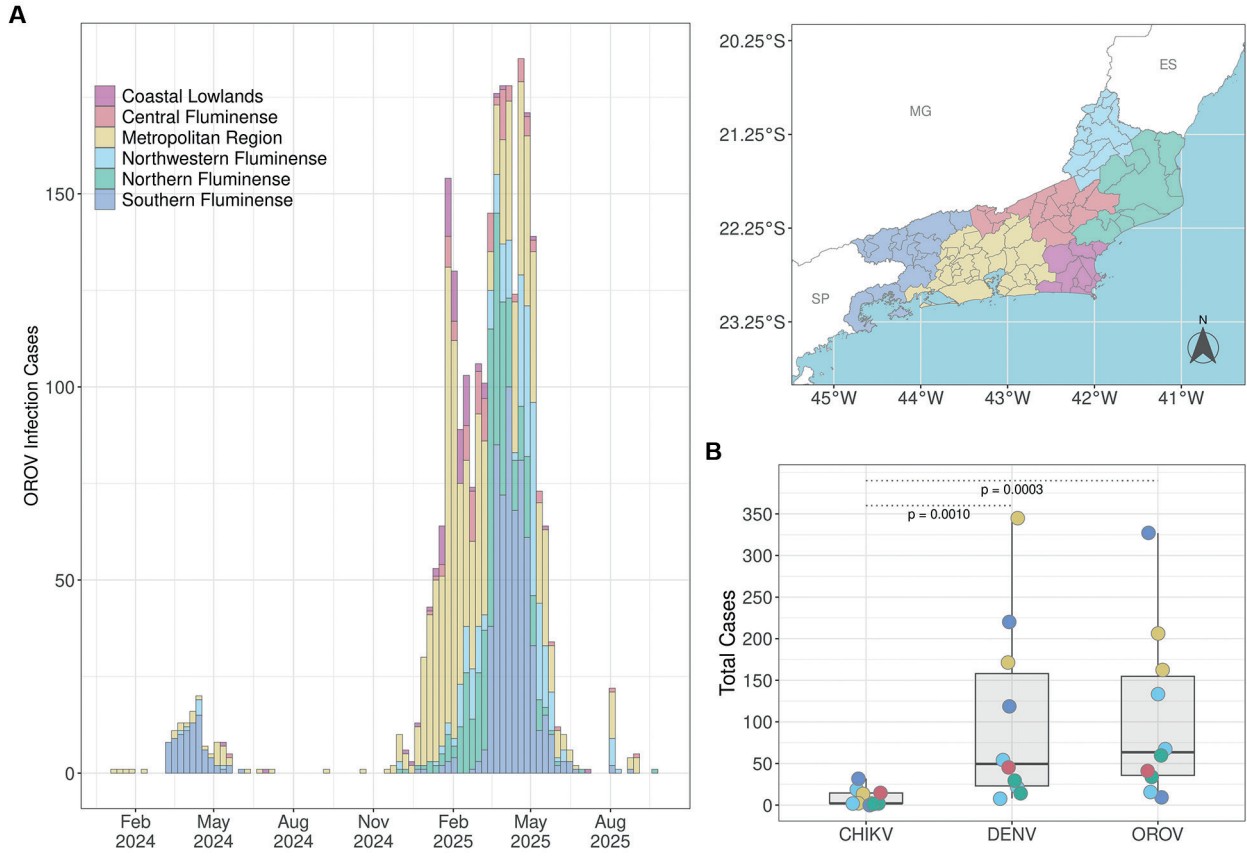

Fig. 1: spatial and temporal distribution of confirmed Oropouche virus (OROV) infection cases in Rio de Janeiro State, 2024-2025. (A) Temporal distribution of confirmed OROV cases, aggregated by epidemiological week (bars), and concurrently stratified by administrative region of infection within Rio de Janeiro State for the 2024-2025 period. Geographical regions depicted on the state map (right) are indicated by colour (inset legend). (B) Total cases of Chikungunya (CHIKV), Dengue (DENV), and OROV infection in the Rio de Janeiro municipalities with the highest Oropouche incidence (> 150 cases/100,000 inhabitants). Points are coloured by state region of municipality location, following the previously established colour scheme. Significance of pairwise comparisons (Mann-Whitney tests) is indicated above each pair of boxplots.

comparable to dengue (n = 103) and significantly higher (p < 0.001) than Chikungunya (n = 9) (Fig. 1B).

Statistical analysis of municipalities with OROV detection (Fig. 2A) revealed that case number was moderately correlated with the forest area [r = 0.50; 95% confidence interval (CI): 0.30-0.66, p < 0.0001], but was not significant correlated (p > 0.05) with either urbanised or grasslands area (Fig. 2B-D). Analysis of the 10 state's top agricultural commodities revealed weak positive correlations between OROV case numbers and the harvesting area of banana (r = 0.36; 95% CI: 0.13-0.55, p = 0.0018) and cassava (r = 0.29; 95% CI: 0.06-0.49, p = 0.012) [Fig. 2E, G and Supplementary data (Figure)]. However, banana and cassava harvesting area was also correlated with forest area (p < 0.01), indicating that these variables are confounded (Fig. 2F, H). In municipalities with low banana+cassava harvested area (≤ 20 ha, n = 29), a moderate correlation between OROV cases and forest area persisted (r = 0.51; 95% CI: 0.17-0.75, p = 0.0045) (Fig. 2I). Conversely, in municipalities with low forest area (≤ 10,000 ha, n = 33), the correlation between OROV cases and banana/cassava harvesting area was no longer significant

(p > 0.05) (Fig. 2J-K). This suggests that forested environments appear to be a more important factor for OROV spread in Rio de Janeiro than cultivated areas.

To trace the origin of the OROV spreading across Rio de Janeiro State, we sequenced 35 OROV genomes (complete or partial) recovered from March 2024 to February 2025 across 15 municipalities in all five state regions, encompassing areas affected by large (n = 6), medium (n = 4), and small (n = 5) outbreaks (Fig. 3A-B). Most OROV genomes were sampled in the Southern Fluminense (n = 14) and Metropolitan (n = 20) regions. The ML phylogenetic analyses of individual genomic segments confirmed that all new genomes from Rio de Janeiro consistently clustered within the OROV$_{BR-2015-2025}$ lineage (Fig. 3C). Next, a concatenated alignment of genomic segments L, M, and S was generated, comprising all OROV sequences from Rio de Janeiro State generated in this (n = 35) and in a previous (n = 5) study, and sequences representative of major OROV$_{BR-2015-2025}$ Amazonian sub-clades previously described.[6] The new ML analysis indicates that most OROV sequences from Rio de Janeiro (n = 36) were nested within the OROV$_{AMACRO-II}$ sub-clade, while four sequences branched within the OROV$_{AM-II}$ sub-clade (Fig. 3D).

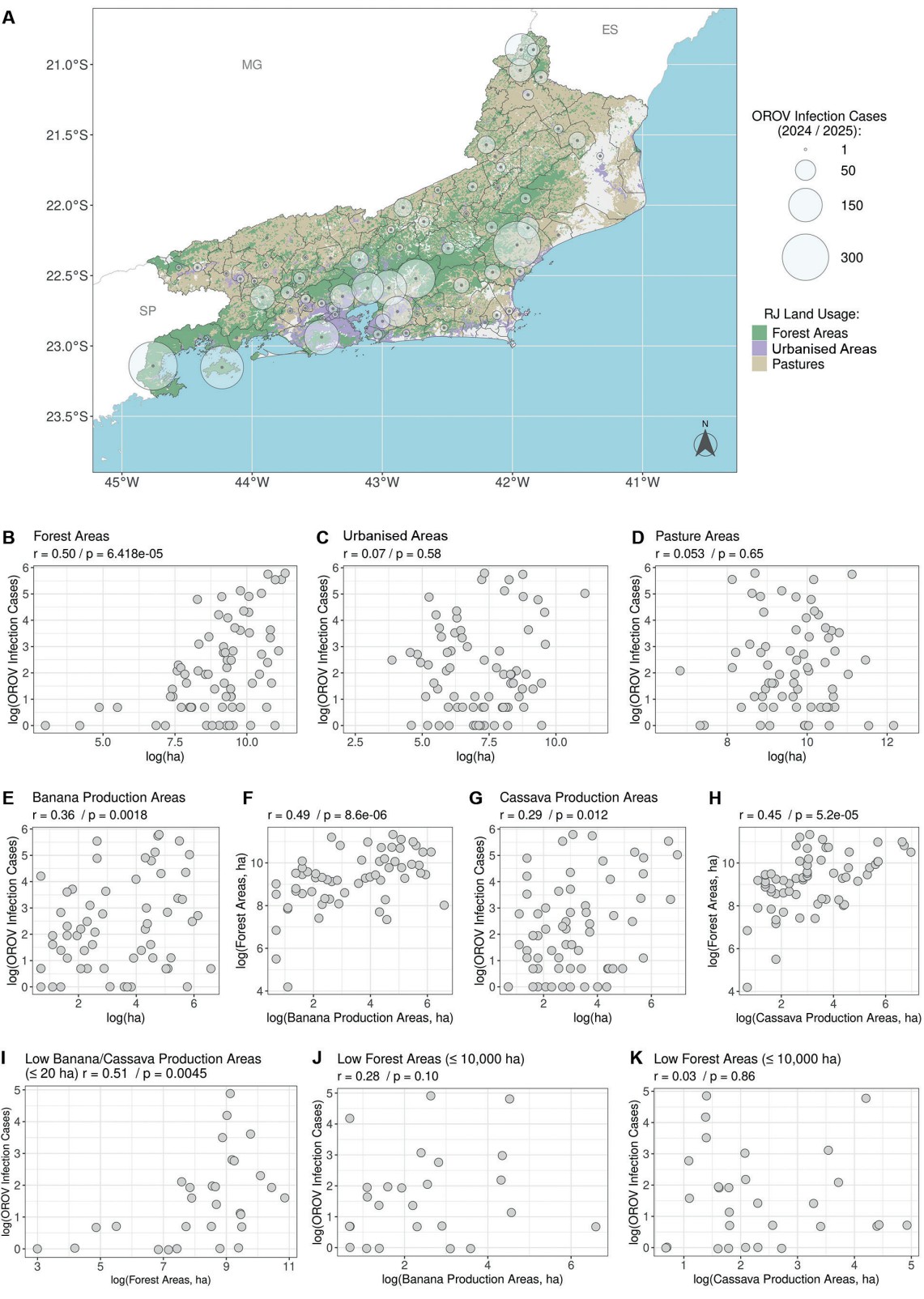

Fig. 2: land use and Oropouche virus (OROV) infection burden in Rio de Janeiro. (A) Cumulative spatial distribution of confirmed OROV cases by municipality over the surveillance period. Circle diameter is proportional to total cases per municipality. Background shading denotes land-cover classes (forest, urban, pasture) coded in the right-hand legend. (B-D) Spearman correlations between total OROV cases per municipality and land-cover area: (B) forest, (C) urban, (D) pasture. (E-F) Spearman correlations between total municipal area dedicated to banana production and (E) total OROV cases, (F) forest area. (G-H) Spearman correlations between total municipal area dedicated to cassava production and (G) total OROV cases, (H) forest area. (I) Spearman correlation between total OROV cases and forest area, restricted to municipalities with low cultivated area of banana and cassava (≤ 20 ha). (J-K) Spearman correlations between total OROV cases and cultivated area of (J) banana and (K) cassava, restricted to municipalities with low forest cover (≤ 10,000 ha). All points represent municipalities; Spearman's ρ and p-values are reported within panels (two-sided).

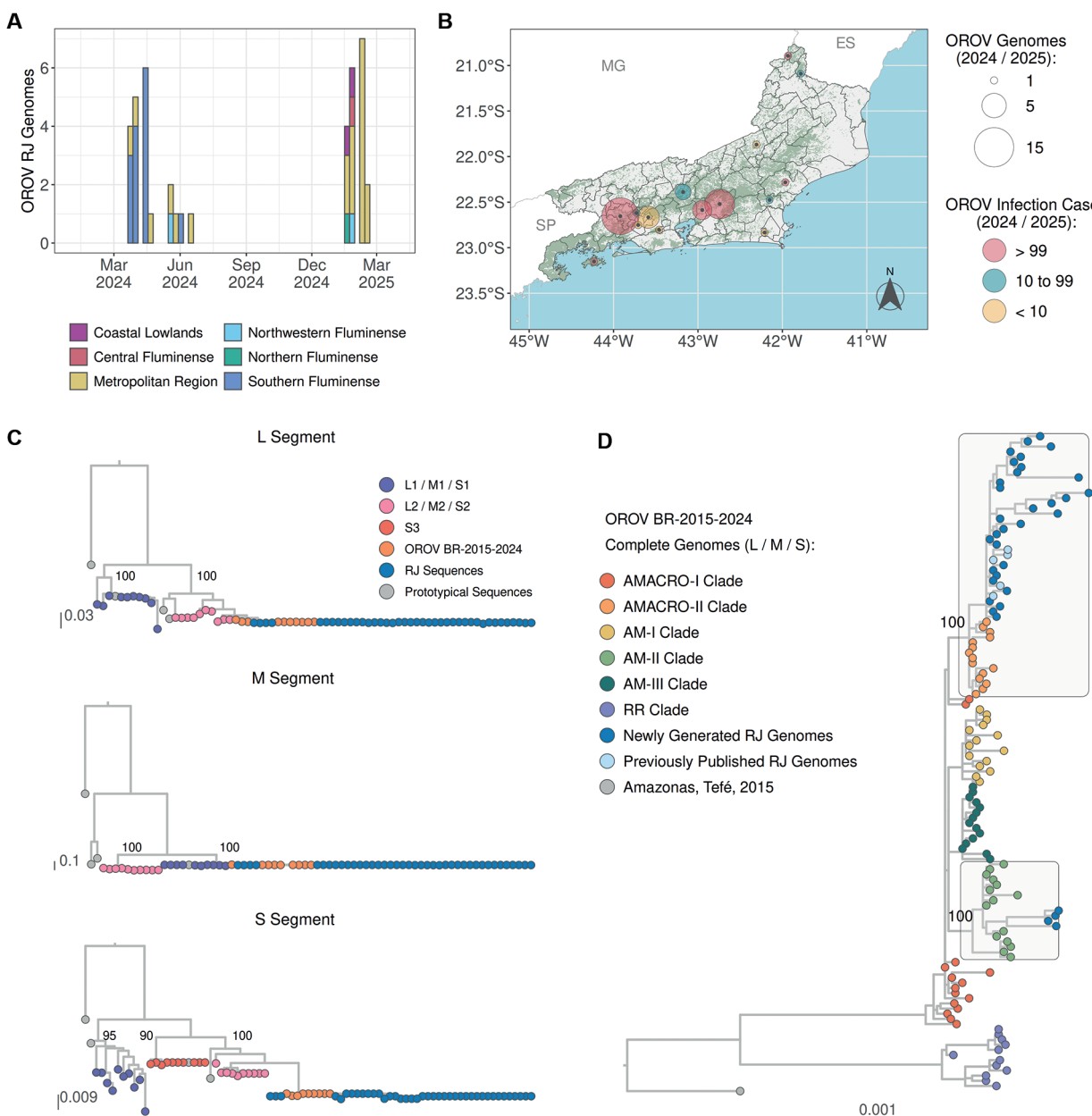

Fig. 3: Oropouche virus (OROV) genomes from Rio de Janeiro State, 2024-2025. (A) Temporal distribution of OROV genomes from Rio de Janeiro (*n* = 40), aggregated by epidemiological week (represented by bars) for the 2024-2025 surveillance period and concurrently stratified by the administrative region where the infection likely occurred. (B) Cumulative spatial distribution map displaying the locations of sampled OROV genomes at the municipal level throughout the surveillance period. The diameter of the circles represents the total number of genomes sequenced per municipality and their colours reflect the magnitude of the underlying Oropouche epidemic. Both encodings follow the legend on the right of the panel. Areas shaded in green indicate regions of forest coverage. (C) Genotyping of OROV L, M, and S genome segments via maximum-likelihood (ML) phylogenetic trees. Sequences were coloured according to their lineage/clade as indicated in the key located in the upper right corner. Sequences from Rio de Janeiro are indicated in blue and prototypical sequences (OROV, Perdões virus, Madre de Dios virus, and Iquitos virus) in grey. Branch support values, calculated using the approximate Likelihood Ratio Test (aLRT), are annotated on key nodes defining the main lineages. (D) ML phylogenetic tree of concatenated OROV segments. The dataset includes representatives of all clades previously identified within the OROV$_{BR-2019-2024}$ lineage. Sub-clades containing genomes from Rio de Janeiro are highlighted and annotated with their statistical support values.

To model the OROV diffusion process in Rio de Janeiro, we thus included in subsequent analyses the earliest known sequences of the OROV$_{BR-2015-2025}$ clade detected in 2015 and 2020, all OROV sequences from the Amazonian region classified within sub-clades OROV$_{AMACRO-I}$, OROV$_{AMACRO-II}$, and OROV$_{AM-II}$, and all sequences from non-Amazonian states nested within those Amazonian sub-clades.[4,6] The correlation between genetic divergence and sampling time supports a significant (p < 0.05) temporal structure for the selected OROV dataset (Fig. 4A). Bayesian discrete phylogeographic analysis revealed a clear geographic division: all sequences from

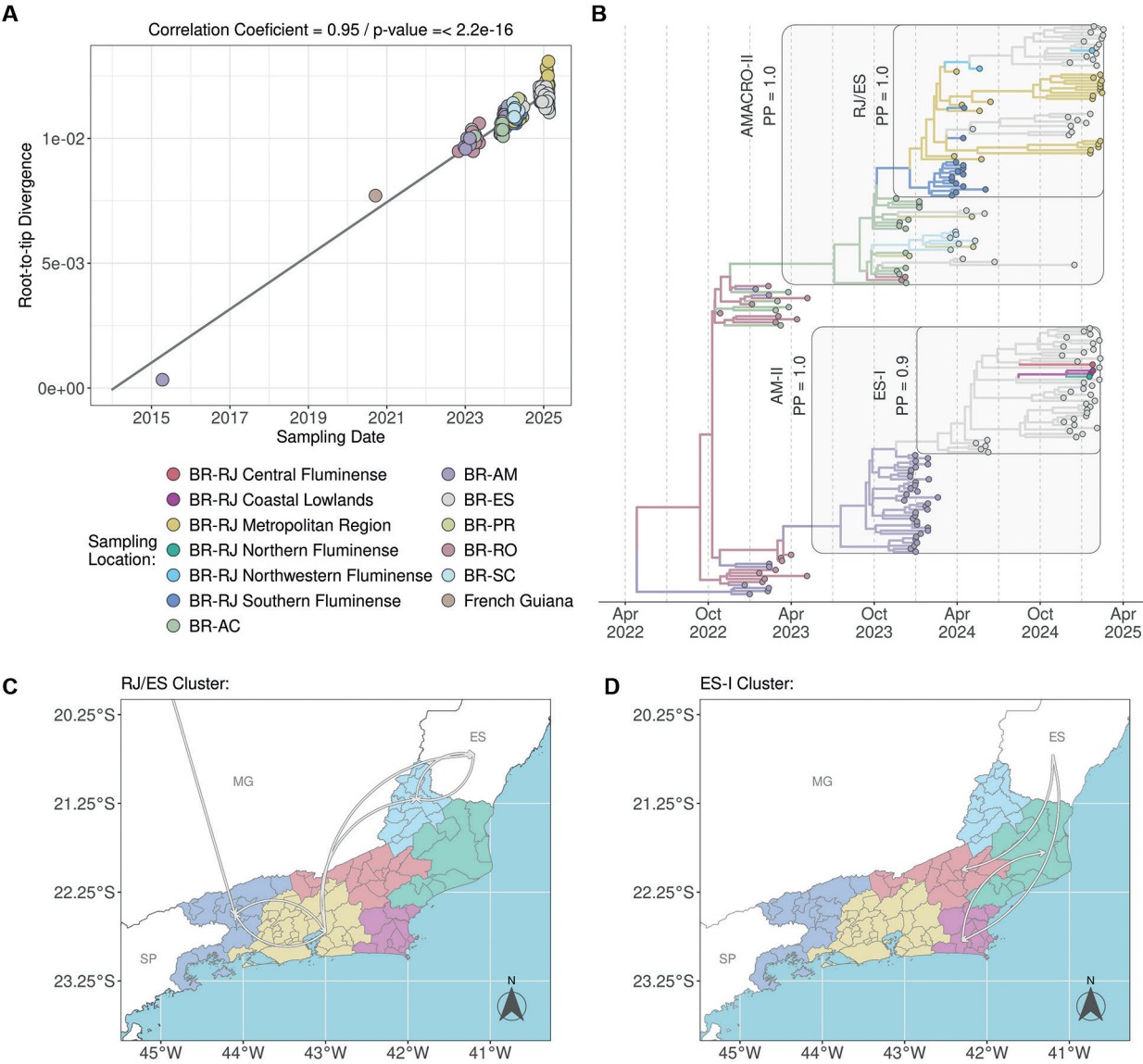

Fig. 4: discrete phylogeographic analysis of Oropouche virus (OROV) dissemination in Rio de Janeiro State, 2024-2025. (A) Evaluation of temporal signal through a linear regression of root-to-tip genetic divergence versus sampling date for selected OROV dataset (n = 191) comprising the oldest OROV$_{BR-2015-2024}$ lineage genomes (Amazonas State, 2015; French Guiana, 2020) and the complete AMACRO-I (n = 28), AMACRO-II (n = 86), and AM-II (n = 75) clades. Data points are colour-coded according to sampling location, corresponding to the legend at the panel's base. (B) Time-scaled Bayesian phylogeographic reconstruction inferred using the previously described genomic dataset. Branches are coloured based on the inferred discrete ancestral location state, employing the colour scheme from panel (A). Clades of interest are highlighted (light grey background) and annotated with established nomenclature and *posterior probability* (PP) support values. The basal 2015 and 2020 sequences were excluded from the graphical display for improved clarity. (C-D) Schematic illustrations summarising the inferred viral migration pathways involving Rio de Janeiro State, as deduced from the phylogeographic analysis presented in (B). Panel (C) delineates migration dynamics associated with the RJ/ES cluster, whereas panel (D) illustrates those pertinent to the ES-I cluster. Both maps employ the colour scheme for state administrative regions defined in panel (A). BR: Brazil; AC: Acre; AM: Amazonas; ES: Espírito Santo; PR: Paraná; RJ: Rio de Janeiro; RO: Rondônia; RR: Roraima; SC: Santa Catarina; AMACRO: Amazonas (AM) + Acre (AC) + Rondônia (RO).

Metropolitan, Southern Fluminense, and Northwest Fluminense regions grouped within the OROV$_{RJ/ES}$ subclade (nested in OROV$_{AMACRO-II}$), whereas those from the Coastal Lowlands, Northern Fluminense, and Central Fluminense regions belonged to the OROV$_{ES-I}$ sub-clade (nested in OROV$_{AM-II}$) (Fig. 4B).

This analysis also indicates that the OROV$_{RJ/ES}$ subclade was likely introduced from the Acre State [posteri-

or state probability (PSP) = 1] into the Southern Fluminense region (PSP = 0.67) around 18 January 2024 (95% highest posterior density (HPD): 16 December 2023 - 13 February 2024). It subsequently spread to the Metropolitan region on 07 February 2024 (95% HPD: 09 January 2024 - 29 February 2024), and from there to the Northwestern Fluminense region and to the Espirito Santo State, before moving back from Espirito Santo into the

Northwestern Fluminense region (Fig. 4C). Meanwhile, the OROV$_{ES-I}$ sub-clade was likely introduced from Espírito Santo State (PSP = 1.0) into the Coastal Lowlands region (PSP = 0.91) around 21 November 2024 (95% HPD: 12 October 2024 - 28 December 2024), and later spread from the Coastal Lowlands to the Northern Fluminense region (Fig. 4D). We also detected a second introduction of the OROV$_{ES-I}$ sub-clade from Espírito Santo into the Central Fluminense region, though we found no evidence of subsequent local dissemination.

We next applied a continuous spatial diffusion model with non-homogeneous dispersion rates to reconstruct the fine-scaled dispersion of the OROV$_{RJ/ES}$ sub-clade. The epicentre of the OROV$_{RJ/ES}$ sub-clade was placed in the municipality of Piraí, in the Southern Fluminense region, from where the virus moved northwards into the Metropolitan and Northwest Fluminense regions in the first half of 2024 and remained circulating in the Metropolitan region during the second half of 2024 (Fig. 5A). Most viral migration events occurred over very-short (< 2 km, 50%) and short (2-10 km, 30%) distances and the average dispersion rate of the OROV$_{RJ/ES}$ sub-clade was estimated at 0.3 km/day (95% HPD: 0.2-0.4 km/day) (Fig. 5B-C). It is interesting to note that during the period of cryptic circulation from July to November 2024, during which no cases were detected, the median OROV dispersion rate dropped from 0.4 km/day to 0.1 km/day (Fig. 5D). Despite this reduction, the virus continued to spread undetected in the Metropolitan region, sustained by at least two independent transmission chains that generated new infections during that period (Fig. 5E).

### DISCUSSION

Our findings revealed that the 2024-2025 OROV epidemic in Rio de Janeiro State was primarily driven by sustained local transmission of the OROV$_{RJ/ES}$ sub-clade introduced in early 2024, alongside multiple independent introductions of the OROV$_{ES-I}$ sub-clade from the neighbouring state of Espírito Santo. The OROV$_{RJ/ES}$ sub-clade fuelled outbreaks in Southern Fluminense, Metropolitan, and Northwestern Fluminense regions, which together account for over 75% of the state's reported OROV cases. Meanwhile, the OROV$_{ES-I}$ sub-clade likely drove outbreaks in the Coastal Lowlands and Northern Fluminense regions.

During 2024-2025, OROV transmission expanded from its endemic Amazon region into the Atlantic Forest biome, where it established autochthonous transmission cycles. This phenomenon was reported across several municipalities of the Southeastern (Espírito Santo, Minas Gerais, and Rio de Janeiro), Northeastern (Alagoas, Bahia, Ceará, and Pernambuco) and Southern (Santa Catarina and Paraná) states of Brazil.[6,7,34-39] Those studies revealed that OROV outbreaks in the Atlantic Forest biome were initially driven by repeated introductions of the OROV$_{BR-2015-2025}$ clade from the Acre/Rondônia and Amazonas states. Our study, however, detected recurrent extra-Amazonian viral migrations of the OROV$_{RJ/ES}$ and OROV$_{ES-I}$ sub-clades between Rio de Janeiro and Espírito Santo states. A recent study also detected extra-Amazonian viral migrations of the OROV$_{RJ/ES}$ sub-clade from Rio de Janeiro to Pernambuco, and of the OROV$_{PE-I}$ sub-clade from Pernambuco to Paraíba and Sergipe states.[39] These findings indicate that long-distance movement of infected individuals between Southeastern and Northeastern states also contributes to sustaining extra-Amazonian OROV transmission networks in Brazil.

The OROV$_{RJ/ES}$ sub-clade spreads primarily over short distances (< 10 km) among contiguous forested municipalities of Rio de Janeiro, likely driven by the dispersal of local infected vectors. Indeed, the average dispersion rate of the OROV$_{RJ/ES}$ sub-clade was estimated at 0.3 km/day (95% HPD: 0.2 - 0.4 km/day), consistent with known active flight ranges of *Culicoides* spp. Vectors.[40,41,42] This mean dispersion rate was lower than those estimated for sub-clades circulating in the Amazonas State (0.66 km/day) and in Acre, Rondônia, and Roraima states (1.00 km/day), but the proportion of short-distance OROV movements (< 10 km) was comparable between Rio de Janeiro (80%) and Northern (70%) states.[4] This indicates a similar pattern of OROV dispersal across both the Amazonian and Atlantic Forest biomes, and further suggests that the mean viral dispersion rate is likely influenced by vector flight range and the geographic distribution range of viral sub-clades, rather than any possible phenotypic differences among OROV sub-lineages.

The exceptional unprecedented expansion of OROV raises questions about potential shifts in vector-virus interactions.[43] Entomological studies conducted in 2024 detected OROV RNA in *Culex quinquefasciatus* and *Aedes albopictus* within an urban setting of the Brazilian Amazon, as well as in *Cx. quinquefasciatus* in Cuba,[44] suggesting the possible involvement of alternative vectors in the transmission of current OROV strains. Other studies, however, indicate that those mosquitoes species are not efficient vectors for transmission of either historic or current OROV isolates.[45-52] Instead, a recent study shows that the current OROV strain has a shorter extrinsic incubation period and higher transmission potential in *C. sonorensis* than the historic one[53] Our findings show a correlation between OROV cases and the extent of municipal forest area, suggesting that urban vectors were likely not a primary driver of the epidemic in Rio de Janeiro. We propose that the synergistic effect of increased vector competence in biting midges, permissive climatic and ecologic factors, and high rates of human mobility could contribute to the establishing OROV transmission cycles beyond the endemic region.

OROV transmission outside the Amazon biome is concentrated in small municipalities where ecological conditions likely favour proliferation of biting midges, but the precise ecological niches exploited by OROV, including potential reservoirs, remains unclear. Some studies conducted at the Atlantic forest biome found a weak/moderate positive correlation (r = 0.39-0.55) between the number of OROV cases and the cultivated area of some crops including: banana, coffee, cacao, coconut, and pepper.[6,38] Our findings indicate weak positive correlations between OROV cases and the harvesting area of banana (r = 0.36) and cassava (r = 0.29). However, the cultivated area of these crops was also positively correlated with forest cover (r = 0.46-0.49), which itself

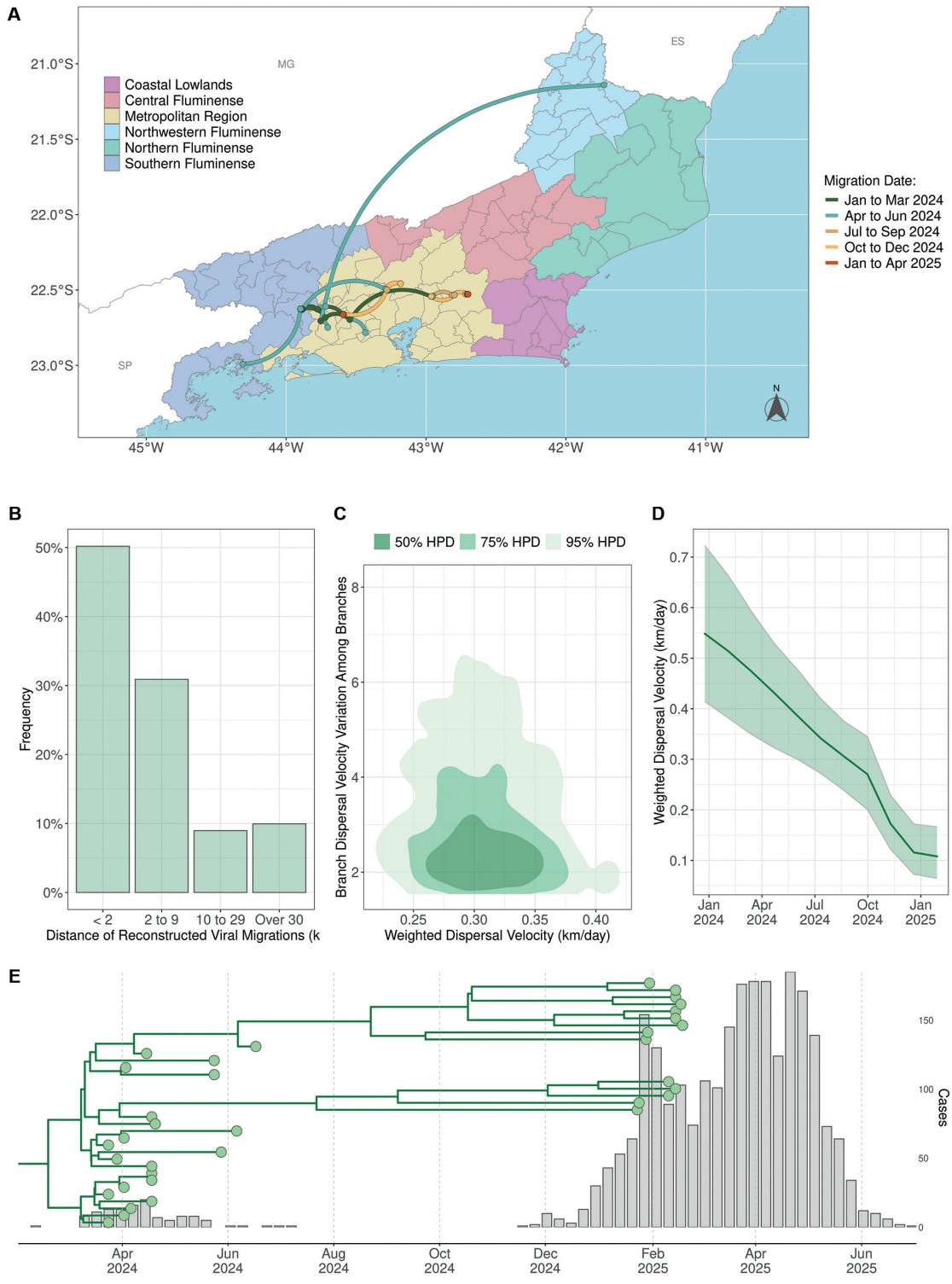

Fig. 5: continuous phylogeographic analysis of Oropouche virus (OROV) dissemination in Rio de Janeiro State, 2024-2025. The panels present results inferred via continuous phylogeographic reconstruction, focusing on OROV genomes from Rio de Janeiro belonging to the RJ/ES cluster (n = 35). (A) Spatial projection of the inferred viral dissemination pathways among the Rio de Janeiro regions. Each curved line represents a corresponding branch from the Bayesian Maximum Clade Credibility tree. Lines are depicted as curved trajectories with a clockwise orientation. The colour of each line segment corresponds to the estimated mean date of that lineage's occurrence, according to the colour key on the panel's right side. (B) Distribution of inferred intra-state migration distances, stratified by distance category: very short (< 2 km), short (2 - 9 km), medium (10 - 29 km), and large (≥ 30 km). (C) Joint posterior density estimate (represented by a 2D kernel density plot) for the weighted dispersal velocity (km/day, x-axis) versus the among-branch variation in this velocity (y-axis). Shaded areas denote the 50% (dark grey), 75% (medium grey), and 95% (light grey) highest posterior density (HPD) intervals for the joint parameter estimates. (D) Temporal evolution trajectory of the estimated weighted dispersal velocity (km/day). The thicker line indicates the median posterior estimate, with the corresponding 95% HPD interval represented by the light green shaded area. (E) Time-scaled maximum clade credibility (MCC) tree of the RJ/ES cluster, overlaid with total OROV case counts in Rio de Janeiro during 2024-2025.

was positively correlated with OROV cases (r = 0.50). A recent study that uses advanced statistical models to handle complex ecological interrelationships, indicates that evergreen trees' land cover and proximity to human settlements contribute to the spread of OROV in Latin America.[54] This supports that establishment of agricultural or residential areas near forests may constitute a high-risk environment for OROV transmission among humans in non-endemic regions.

The transmission of OROV in municipalities with periurban and agricultural areas adjacent to well-preserved Atlantic Forest remnants creates an ecological interface that may intensify contact between humans, vectors, and reservoirs, thereby facilitating the virus exchange between urban and sylvatic transmission cycle. In 2024, a significant gap of 4.5 months with no reported human OROV cases was observed in Rio de Janeiro, spanning the dry season from early July to mid-November. Notably, our phylogeographic analysis suggests that from July to November 2024, the rate of spread of the $OROV_{RJ/ES}$ sub-clade was substantially reduced from 0.4 to 0.1 km/day, but the virus continued to move and seed new infections through contiguous forest areas of the Metropolitan region. We hypothesise that Atlantic Forest remnants in Rio de Janeiro's Metropolitan region may have served as a dry-season refuge for the $OROV_{RJ/ES}$ sub-clade, sustaining a cryptic sylvatic transmission cycle characterised by a slow viral dispersion rate. This sylvatic reservoir, in turn, could have contributed to new urban transmission cycles in municipalities near forested areas during the following rainy season in 2025.

Our study has three major limitations. First, the estimated origin of the $OROV_{RJ/ES}$ sub-clade around mid-January 2024 should be interpreted with caution due to: (i) absence of genomic data before March 2024, and (ii) the probable underreporting of early cases in official surveillance because of possible clinical misdiagnosis.[55] Second, our viral genomic dataset covered 10 of the 26 municipalities in Rio de Janeiro State that reported medium or large OROV outbreaks. While the main conclusion of multi-year persistence for the $OROV_{RJ/ES}$ sub-clade is supported by the available data, the precise dispersal routes of the $OROV_{RJ/ES}$ and $OROV_{ES-I}$ sub-clades may have been influenced by the lack or the very low number of viral genomes from several heavily affected municipalities. A third limitation is the lack of geolocalised data from human infections, which precluded a fine-scale spatial analysis of the relationship between OROV cases and forested area. Future studies should prioritise the systematic collection of geolocated data, coupled with the genomic analysis of representative viral samples from diverse municipalities across Rio de Janeiro.

In summary, these results highlight the potential for sustained autochthonous OROV transmission in Atlantic Forest regions of Rio de Janeiro State, and suggest the possibility of establishing an endemic transmission cycle beyond the Amazon region. These findings underscore the importance of routinely including OROV in differential diagnoses of acute febrile illnesses in all Brazilian states and the need to implement a One Health OROV surveillance approach focused on the vectors and potential vertebrate hosts in the Atlantic forest biome. The successful transmission of OROV in Rio de Janeiro also raises concerns about the potential expansion of this arbovirus beyond South America.

## ACKNOWLEDGEMENTS

To the Next Generation Sequencing Platform (RPT01J) - Technology Platforms Network/VPPCB - FIOCRUZ, and the Respiratory, Exanthematous, Enterovirus and Viral Emergencies Laboratory of the Oswaldo Cruz Institute (IOC/Fiocruz) for their support in building and reading the libraries, and to collaborators Solange Regina da Conceição and Ronaldo Lapa Lopes for their technical support.

## AUTHORS' CONTRIBUTION

IA - conceptual design, phylogenetic data analysis, manuscript's original draft; FBN - conceptual design, retrieval and analysis of clinical data in RJ State, generation and assembly of genomic data; CO - retrieval and analysis of clinical data in RJ State, generation and assembly of genomic data; PCS, OL, MB, CLM, MB, AB, LSV, ACC and FBC - retrieval and analysis of clinical and epidemiological data in RJ State; CGCM, VZD, TJS, FDG, GCM, JZN, RR-R and ED - retrieval and analysis of clinical and epidemiological data in ES State; GC - conceptual design, retrieval and analysis of clinical data in RJ State; FGN - conceptual design, phylogenetic data analysis, funding acquisition, project administration; PB - conceptual design, ethical approval, retrieval and analysis of clinical data in RJ State, funding acquisition, project administration; AMBF - conceptual design, ethical approval, funding acquisition, project administration; GB - conceptual design, phylogenetic data analysis, manuscript's original draft. All authors have revised the final version of the manuscript. The funders had no role in study design, data collection and analysis, decision to publish, or preparation of the manuscript.

## DATA AVAILABILITY

Consensus sequences of OROV genomes generated in this study are available at GISAID (https://doi.org/10.55876/gis8.250523gx).

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

# OPEN PEER REVIEW

Memórias do IOC thanks the anonymous reviewers for their contribution to the peer review of this work.

## FIRST REVIEW ROUND

REVIEWERS' COMMENTS

### REVIEWER #1

In the article by Arantes et al., "Sustained Oropouche virus transmission in Rio de Janeiro's Atlantic forest: genomic evidence over a two-year period," the authors characterize OROV dissemination in Rio de Janeiro state during 2024–2025. An outbreak began in Southern Fluminense in 2024 (n = 151) and, by April 30, 2025, had expanded to 1,075 reported cases, with widespread transmission in rural areas of the Metropolitan and Northern Fluminense regions. To investigate this, the authors applied genomic surveillance and phylodynamic methods to a dataset of 40 OROV genomes, including 35 newly generated sequences (16 from 2024 and 19 from 2025) from clinical samples collected across 15 municipalities in Rio de Janeiro. The discrete phylogeographic analysis inferred two main routes of OROV dissemination in Brazil. The OROVRJ/ES clade was introduced into Southern Fluminense around January 2024, subsequently spreading to the Metropolitan region, Northwest Fluminense, and Espírito Santo. The OROVES-I clade originated in Espírito Santo and was introduced into Central Fluminense and the Coastal Lowlands, later reaching Northern Fluminense. Under a continuous diffusion model, OROVRJ/ES dispersal occurred predominantly via very short and short-distance (>10 km, 80%) movements, with a mean rate of 0.3 km/day (95% CI: 0.2–0.4). This rate declined from 0.5 km/day in January 2024 to 0.1 km/day by January 2025. The authors conclude that the OROVRJ/ES sub-clade persisted cryptically in the Metropolitan region during late 2024, subsequently seeding new outbreaks in early 2025. These findings underscore the potential for sustained OROV transmission across multiple years in Rio de Janeiro's Atlantic Forest and highlight the risk of extra-Amazonian expansion. Overall, this manuscript is an important contribution and can be accepted after minor revisions.

Minor revisions:
1) In the abstract, adjust the placement of the sentence fragment "(16 from 2024 and 19 from 2025)" to read as follows: "To investigate this, the authors applied genomic surveillance and phylodynamic methods to a dataset of 40 OROV genomes, including 35 newly generated sequences (16 from 2024 and 19 from 2025) from clinical samples collected across 15 municipalities in Rio de Janeiro."
2) Add in the Introduction that OROV has a segmented genome, enabling reassortment, which has facilitated its spread beyond the Amazon.
3) In the Materials and Methods section, the authors should provide a detailed description of the primers and assay conditions used for the RT-qPCR.
4) Page 7, line 154: the word "using" appears duplicated.
5) In the Results section, please specify how many of the 35 novel OROV genomes are complete and how many are partial (currently reported as "A total of 35 novel OROV genomes [complete or partial]").
6) In the discussion need many improvements:
6.1) Cite recent studies documenting the spread of OROV outside the Amazon, with confirmed cases in Espírito Santo, Rio de Janeiro, Minas Gerais, Mato Grosso, Bahia, and Santa Catarina;
6.1.1) Godinho et al., 2025. Insights into the expansion of Oropouche virus in Brazil: epidemiological and environmental aspects. Exp Biol Med (Maywood). 250:10647. doi: 10.3389/ebm.2025.10647.
6.1.2) Medeiros et al., 2025. Case Series of Adverse Pregnancy Outcomes Associated with Oropouche Virus Infection. Viruses. 17(6):816. doi: 10.3390/v17060816.
6.1.3) Mendonça et al., 2025. Oropouche orthobunyavirus in Urban Mosquitoes: Vector Competence, Coinfection, and Immune System Activation in Aedes aegypti. Viruses. 17(4):492. doi: 10.3390/v17040492.
6.1.4) Schwartz, 2025. Novel Reassortants of Oropouche Virus (OROV) Are Causing Maternal-Fetal Infection During Pregnancy, Stillbirth, Congenital Microcephaly and Malformation Syndromes. Genes (Basel). 16(1):87. doi: 10.3390/genes16010087.
6.2) Relate the Rio de Janeiro outbreak to the national scenario, demonstrating that it is not an isolated event but part of a broader trend of virus spread;
6.3) Discuss the importance of ecological and environmental factors. For example, (i) highlight the role of the remaining forest cover in the Atlantic Forest and the proximity of rural/periurban areas in maintaining transmission; and (ii) discuss the possibility of multiple vectors being involved (Culicoides vs. mosquitoes), which is still under debate.
6.4) Add study limitations: (i) sample restricted to the state of Rio de Janeiro and the 2024–2025 period; (ii) underrepresentation of asymptomatic or undiagnosed cases; (iii) reliance on official surveillance data, subject to underreporting; (iv) possible bias in spatial representation (not all affected municipalities had genomes included).

6.5) Add future perspectives: (i) importance of strengthening genomic surveillance to monitor potential reassortments, since OROV has a segmented genome; (ii) need to integrate human, vector, and wild reservoir surveillance (One Health approach); (iii) development of sensitive differential diagnostic tests for OROV, especially in regions endemic for dengue, ZIKV, and CHIKV, to avoid misdiagnoses; (iv) putting the study into an international perspective: expansion beyond the Amazon reinforces that neglected arboviruses can become national and even regional challenges.

7. Page 17, line 366: Please correct the link to the epidemiological dashboard — there is an unintended space (gap) in the URL.

8. Always use "Rio de Janeiro state" (or "the state of Rio de Janeiro") consistently. In some places, it simply appears as "Rio de Janeiro," which can be confusing for the city.

## AUTHORS' RESPONSE TO THE REVIEWERS

REVIEWER COMMENTS:

Reviewer: 1

In the article by Arantes et al., "Sustained Oropouche virus transmission in Rio de Janeiro's Atlantic forest: genomic evidence over a two-year period," the authors characterize OROV dissemination in Rio de Janeiro state during 2024–2025. An outbreak began in Southern Fluminense in 2024 (n = 151) and, by April 30, 2025, had expanded to 1,075 reported cases, with widespread transmission in rural areas of the Metropolitan and Northern Fluminense regions. To investigate this, the authors applied genomic surveillance and phylodynamic methods to a dataset of 40 OROV genomes, including 35 newly generated sequences (16 from 2024 and 19 from 2025) from clinical samples collected across 15 municipalities in Rio de Janeiro. The discrete phylogeographic analysis inferred two main routes of OROV dissemination in Brazil. The OROVRJ/ES clade was introduced into Southern Fluminense around January 2024, subsequently spreading to the Metropolitan region, Northwest Fluminense, and Espírito Santo. The OROVES-I clade originated in Espírito Santo and was introduced into Central Fluminense and the Coastal Lowlands, later reaching Northern Fluminense. Under a continuous diffusion model, OROVRJ/ES dispersal occurred predominantly via very short and short-distance (>10 km, 80%) movements, with a mean rate of 0.3 km/day (95% CI: 0.2–0.4). This rate declined from 0.5 km/day in January 2024 to 0.1 km/day by January 2025. The authors conclude that the OROVRJ/ES sub-clade persisted cryptically in the Metropolitan region during late 2024, subsequently seeding new outbreaks in early 2025. These findings underscore the potential for sustained OROV transmission across multiple years in Rio de Janeiro's Atlantic Forest and highlight the risk of extra-Amazonian expansion. Overall, this manuscript is an important contribution and can be accepted after minor revisions.

Minor Revisions:

1) In the abstract, adjust the placement of the sentence fragment "(16 from 2024 and 19 from 2025)" to read as follows: "To investigate this, the authors applied genomic surveillance and phylodynamic methods to a dataset of 40 OROV genomes, including 35 newly generated sequences (16 from 2024 and 19 from 2025) from clinical samples collected across 15 municipalities in Rio de Janeiro."

Reply: The text was changed accordingly.

2) Add in the Introduction that OROV has a segmented genome, enabling reassortment, which has facilitated its spread beyond the Amazon.

Reply: Regarding the potential fitness implications of the reassortment that gave rise to the M1L2S2 genotype, the literature is conflicted. While some reports describe augmented viral kinetics, pathogenicity, and immune escape (DOI: 10.1016/S1473-3099(24)00619-4), another study (DOI: 10.1016/ S1473-3099(25)00110-0) integrating serology, antigenic cartography, and in-vitro phenotyping found that the 2023–2024 outbreak strain did not outperform historical or contemporary OROV strains in replication kinetics or plaque size, and that strong antigenic differentiation is evident chiefly between OROV and its M-segment glycoprotein reassortants (e.g., IQTV, MDDV), not among OROV strains themselves. Results of the later study were considered by us at the moment as more reliable and are cited in the discussion section of the manuscript.

3) In the Materials and Methods section, the authors should provide a detailed description of the primers and assay conditions used for the RT-qPCR.

Reply: The Methodology section was updated to include the reference of the primers and assay conditions used in the RT-qPCR reaction (DOI: 10.1590/0074-02760160062)

4) Page 7, line 154: the word "using" appears duplicated.

Reply: The text was changed accordingly.

5) In the Results section, please specify how many of the 35 novel OROV genomes are complete and how many are partial (currently reported as "A total of 35 novel OROV genomes [complete or partial]").

Reply: A supplementary table was added to the manuscript containing the coverage of the tree segments for the genomic sequence samples in Rio de Janeiro.

6) In the discussion need many improvements:

6.1) Cite recent studies documenting the spread of OROV outside the Amazon, with confirmed cases in Espírito Santo, Rio de Janeiro, Minas Gerais, Mato Grosso, Bahia, and Santa Catarina;

6.1.1) Godinho et al., 2025. Insights into the expansion of Oropouche virus in Brazil: epidemiological and environmental aspects. Exp Biol Med (Maywood). 250:10647. doi: 10.3389/ebm.2025.10647.

6.1.2) Medeiros et al., 2025. Case Series of Adverse Pregnancy Outcomes Associated with Oropouche Virus Infection. Viruses. 17(6):816. doi: 10.3390/v17060816.

6.1.3) Mendonça et al., 2025. Oropouche orthobunyavirus in Urban Mosquitoes: Vector Competence, Coinfection, and Immune System Activation in Aedes aegypti. Viruses. 17(4):492. doi: 10.3390/v17040492.

6.1.4) Schwartz, 2025. Novel Reassortants of Oropouche Virus (OROV) Are Causing Maternal-Fetal Infection During Pregnancy, Stillbirth, Congenital Microcephaly and Malformation Syndromes. Genes (Basel). 16(1):87. doi: 10.3390/genes16010087.

Reply: We thank the reviewer for the suggestion to expand the Discussion section and for pointing us to the relevant literature. The manuscript was initially conceived as a concise report, which led to a more focused discussion. We agree that a more comprehensive discussion significantly strengthens the paper, and we have now expanded it accordingly, incorporating the suggested studies."

6.2) Relate the Rio de Janeiro outbreak to the national scenario, demonstrating that it is not an isolated event but part of a broader trend of virus spread;

Reply: The Discussion section of the manuscript was updated to reflect the national picture of sustained autochthonous Oropouche transmission across multiple Brazilian states.

6.3) Discuss the importance of ecological and environmental factors. For example, (i) highlight the role of the remaining forest cover in the Atlantic Forest and the proximity of rural/periurban areas in maintaining transmission; and (ii) discuss the possibility of multiple vectors being involved (Culicoides vs. mosquitoes), which is still under debate.

Reply: To better understand the role of ecological factors, we incorporated new analyses examining associations between Oropouche burden and land-use patterns in the different municipalities of the Rio de Janeiro state. The results, detailed in the Results section, guide our discussion on the potential vectors facilitating OROV dissemination in Rio de Janeiro.

6.4) Add study limitations:

(i) sample restricted to the state of Rio de Janeiro and the 2024–2025 period;

(ii) underrepresentation of asymptomatic or undiagnosed cases;

(iii) reliance on official surveillance data, subject to underreporting;

(iv) possible bias in spatial representation (not all affected municipalities had genomes included).

Reply: The Discussion section was updated based on the reviewer's suggestions.

6.5) Add future perspectives:

(i) importance of strengthening genomic surveillance to monitor potential reassortments, since OROV has a segmented genome;

(ii) need to integrate human, vector, and wild reservoir surveillance (One Health approach);

(iii) development of sensitive differential diagnostic tests for OROV, especially in regions endemic for dengue, ZIKV, and CHIKV, to avoid misdiagnoses;

(iv) putting the study into an international perspective: expansion beyond the Amazon reinforces that neglected arboviruses can become national and even regional challenges.

Reply: The Discussion section was updated based on the reviewer's suggestions.

7. Page 17, line 366: Please correct the link to the epidemiological dashboard — there is an unintended space (gap) in the URL.

Reply: The text was changed accordingly.

8. Always use "Rio de Janeiro state" (or "the state of Rio de Janeiro") consistently. In some places, it simply appears as "Rio de Janeiro," which can be confusing for the city.

Reply: The text was changed accordingly.

## SECOND REVIEW ROUND

REVIEWERS' COMMENTS

**REVIEWER #1**

The manuscript meets all the criteria requested for evaluation. Moreover, the authors have thoroughly addressed all reviewers' comments and incorporated all necessary revisions. The manuscript is now suitable for acceptance in MIOC.

