## [Reviewer Report · FIRST REVIEW ROUND - REVIEWERS COMMENTS]

## REVIEWER #1

In the article by Arantes et al., “Sustained Oropouche virus transmission in Rio de Janeiro’s Atlantic forest: genomic evidence over a two-year period,” the authors characterize OROV dissemination in Rio de Janeiro state during 2024–2025.

An outbreak began in Southern Fluminense in 2024 (n = 151) and, by April 30, 2025, had expanded to 1,075 reported cases, with widespread transmission in rural areas of the Metropolitan and Northern Fluminense regions.

To investigate this, the authors applied genomic surveillance and phylodynamic methods to a dataset of 40 OROV genomes, including 35 newly generated sequences (16 from 2024 and 19 from 2025) from clinical samples collected across 15 municipalities in Rio de Janeiro.

The discrete phylogeographic analysis inferred two main routes of OROV dissemination in Brazil.

The OROVRJ/ES clade was introduced into Southern Fluminense around January 2024, subsequently spreading to the Metropolitan region, Northwest Fluminense, and Espírito Santo.

The OROVES-I clade originated in Espírito Santo and was introduced into Central Fluminense and the Coastal Lowlands, later reaching Northern Fluminense.

Under a continuous diffusion model, OROVRJ/ES dispersal occurred predominantly via very short and short-distance (>10 km, 80%) movements, with a mean rate of 0.3 km/day (95% CI: 0.2–0.4).

This rate declined from 0.5 km/day in January 2024 to 0.1 km/day by January 2025. The authors conclude that the OROVRJ/ES sub-clade persisted cryptically in the Metropolitan region during late 2024, subsequently seeding new outbreaks in early 2025. These findings underscore the potential for sustained OROV transmission across multiple years in Rio de Janeiro’s Atlantic Forest and highlight the risk of extra-Amazonian expansion.

Overall, this manuscript is an important contribution and can be accepted after minor revisions.

Minor revisions:

1) In the abstract, adjust the placement of the sentence fragment “(16 from 2024 and 19 from 2025)” to read as follows: “To investigate this, the authors applied genomic surveillance and phylodynamic methods to a dataset of 40 OROV genomes, including 35 newly generated sequences (16 from 2024 and 19 from 2025) from clinical samples collected across 15 municipalities in Rio de Janeiro.”

2) Add in the Introduction that OROV has a segmented genome, enabling reassortment, which has facilitated its spread beyond the Amazon.

3) In the Materials and Methods section, the authors should provide a detailed description of the primers and assay conditions used for the RT-qPCR.

4) Page 7, line 154: the word “using” appears duplicated.

5) In the Results section, please specify how many of the 35 novel OROV genomes are complete and how many are partial (currently reported as “A total of 35 novel OROV genomes [complete or partial]”).

6) In the discussion need many improvements:

6.1) Cite recent studies documenting the spread of OROV outside the Amazon, with confirmed cases in Espírito Santo, Rio de Janeiro, Minas Gerais, Mato Grosso, Bahia, and Santa Catarina;

6.1.1) Godinho et al., 2025. Insights into the expansion of Oropouche virus in Brazil: epidemiological and environmental aspects.

Exp Biol Med (Maywood). 250:10647. doi: 10.3389/ebm.2025.10647.

6.1.2) Medeiros et al., 2025. Case Series of Adverse Pregnancy Outcomes Associated with Oropouche Virus Infection. Viruses. 17(6):816. doi: 10.3390/v17060816.

6.1.3) Mendonça et al., 2025. Oropouche orthobunyavirus in Urban Mosquitoes: Vector Competence, Coinfection, and Immune System Activation in Aedes aegypti.

Viruses. 17(4):492. doi: 10.3390/v17040492.

6.1.4) Schwartz, 2025. Novel Reassortants of Oropouche Virus (OROV) Are Causing Maternal-Fetal Infection During Pregnancy, Stillbirth, Congenital Microcephaly and Malformation Syndromes.

Genes (Basel). 16(1):87. doi: 10.3390/genes16010087.

6.2) Relate the Rio de Janeiro outbreak to the national scenario, demonstrating that it is not an isolated event but part of a broader trend of virus spread;

6.3) Discuss the importance of ecological and environmental factors. For example, (i) highlight the role of the remaining forest cover in the Atlantic Forest and the proximity of rural/periurban areas in maintaining transmission;

and (ii) discuss the possibility of multiple vectors being involved (Culicoides vs. mosquitoes), which is still under debate.

6.4) Add study limitations: (i) sample restricted to the state of Rio de Janeiro and the 2024–2025 period;

(ii) underrepresentation of asymptomatic or undiagnosed cases; (iii) reliance on official surveillance data, subject to underreporting;

(iv) possible bias in spatial representation (not all affected municipalities had genomes included).

6.5) Add future perspectives: (i) importance of strengthening genomic surveillance to monitor potential reassortments, since OROV has a segmented genome;

(ii) need to integrate human, vector, and wild reservoir surveillance (One Health approach);

(iii) development of sensitive differential diagnostic tests for OROV, especially in regions endemic for dengue, ZIKV, and CHIKV, to avoid misdiagnoses;

(iv) putting the study into an international perspective: expansion beyond the Amazon reinforces that neglected arboviruses can become national and even regional challenges.

7. Page 17, line 366: Please correct the link to the epidemiological dashboard — there is an unintended space (gap) in the URL.

8. Always use “Rio de Janeiro state” (or “the state of Rio de Janeiro”) consistently.

In some places, it simply appears as “Rio de Janeiro,” which can be confusing for the city.

## AUTHORS’ RESPONSE TO THE REVIEWERS

REVIEWER COMMENTS:

Reviewer: 1

In the article by Arantes et al., “Sustained Oropouche virus transmission in Rio de Janeiro’s Atlantic forest: genomic evidence over a two-year period,” the authors characterize OROV dissemination in Rio de Janeiro state during 2024–2025.

An outbreak began in Southern Fluminense in 2024 (n = 151) and, by April 30, 2025, had expanded to 1,075 reported cases, with widespread transmission in rural areas of the Metropolitan and Northern Fluminense regions.

To investigate this, the authors applied genomic surveillance and phylodynamic methods to a dataset of 40 OROV genomes, including 35 newly generated sequences (16 from 2024 and 19 from 2025) from clinical samples collected across 15 municipalities in Rio de Janeiro.

The discrete phylogeographic analysis inferred two main routes of OROV dissemination in Brazil.

The OROVRJ/ES clade was introduced into Southern Fluminense around January 2024, subsequently spreading to the Metropolitan region, Northwest Fluminense, and Espírito Santo.

The OROVES-I clade originated in Espírito Santo and was introduced into Central Fluminense and the Coastal Lowlands, later reaching Northern Fluminense.

Under a continuous diffusion model, OROVRJ/ES dispersal occurred predominantly via very short and short-distance (>10 km, 80%) movements, with a mean rate of 0.3 km/day (95% CI: 0.2–0.4).

This rate declined from 0.5 km/day in January 2024 to 0.1 km/day by January 2025. The authors conclude that the OROVRJ/ES sub-clade persisted cryptically in the Metropolitan region during late 2024, subsequently seeding new outbreaks in early 2025. These findings underscore the potential for sustained OROV transmission across multiple years in Rio de Janeiro’s Atlantic Forest and highlight the risk of extra-Amazonian expansion.

Overall, this manuscript is an important contribution and can be accepted after minor revisions.

Minor Revisions:

1) In the abstract, adjust the placement of the sentence fragment “(16 from 2024 and 19 from 2025)” to read as follows: “To investigate this, the authors applied genomic surveillance and phylodynamic methods to a dataset of 40 OROV genomes, including 35 newly generated sequences (16 from 2024 and 19 from 2025) from clinical samples collected across 15 municipalities in Rio de Janeiro.”

Reply: The text was changed accordingly.

2) Add in the Introduction that OROV has a segmented genome, enabling reassortment, which has facilitated its spread beyond the Amazon.

Reply: Regarding the potential fitness implications of the reassortment that gave rise to the M1L2S2 genotype, the literature is conflicted.

While some reports describe augmented viral kinetics, pathogenicity, and immune escape (DOI: 10.1016/S1473-3099(24)00619-4), another study (DOI: 10.1016/ S1473-3099(25)00110-0) integrating serology, antigenic cartography, and in-vitro phenotyping found that the 2023–2024 outbreak strain did not outperform historical or contemporary OROV strains in replication kinetics or plaque size, and that strong antigenic differentiation is evident chiefly between OROV and its M-segment glycoprotein reassortants (e.g., IQTV, MDDV), not among OROV strains themselves.

Results of the later study were considered by us at the moment as more reliable and are cited in the discussion section of the manuscript.

3) In the Materials and Methods section, the authors should provide a detailed description of the primers and assay conditions used for the RT-qPCR.

Reply: The Methodology section was updated to include the reference of the primers and assay conditions used in the RT-qPCR reaction (DOI: 10.1590/0074-02760160062)

4) Page 7, line 154: the word “using” appears duplicated.

Reply: The text was changed accordingly.

5) In the Results section, please specify how many of the 35 novel OROV genomes are complete and how many are partial (currently reported as “A total of 35 novel OROV genomes [complete or partial]”).

Reply: A supplementary table was added to the manuscript containing the coverage of the tree segments for the genomic sequence samples in Rio de Janeiro.

6) In the discussion need many improvements:

6.1) Cite recent studies documenting the spread of OROV outside the Amazon, with confirmed cases in Espírito Santo, Rio de Janeiro, Minas Gerais, Mato Grosso, Bahia, and Santa Catarina;

6.1.1) Godinho et al., 2025. Insights into the expansion of Oropouche virus in Brazil: epidemiological and environmental aspects.

Exp Biol Med (Maywood). 250:10647. doi: 10.3389/ebm.2025.10647.

6.1.2) Medeiros et al., 2025. Case Series of Adverse Pregnancy Outcomes Associated with Oropouche Virus Infection. Viruses. 17(6):816. doi: 10.3390/v17060816.

6.1.3) Mendonça et al., 2025. Oropouche orthobunyavirus in Urban Mosquitoes: Vector Competence, Coinfection, and Immune System Activation in Aedes aegypti.

Viruses. 17(4):492. doi: 10.3390/v17040492.

6.1.4) Schwartz, 2025. Novel Reassortants of Oropouche Virus (OROV) Are Causing Maternal-Fetal Infection During Pregnancy, Stillbirth, Congenital Microcephaly and Malformation Syndromes.

Genes (Basel). 16(1):87. doi: 10.3390/genes16010087.

Reply: We thank the reviewer for the suggestion to expand the Discussion section and for pointing us to the relevant literature.

The manuscript was initially conceived as a concise report, which led to a more focused discussion.

We agree that a more comprehensive discussion significantly strengthens the paper, and we have now expanded it accordingly, incorporating the suggested studies.”

6.2) Relate the Rio de Janeiro outbreak to the national scenario, demonstrating that it is not an isolated event but part of a broader trend of virus spread;

Reply: The Discussion section of the manuscript was updated to reflect the national picture of sustained autochthonous Oropouche transmission across multiple Brazilian states.

6.3) Discuss the importance of ecological and environmental factors. For example, (i) highlight the role of the remaining forest cover in the Atlantic Forest and the proximity of rural/periurban areas in maintaining transmission;

and (ii) discuss the possibility of multiple vectors being involved (Culicoides vs. mosquitoes), which is still under debate.

Reply: To better understand the role of ecological factors, we incorporated new analyses examining associations between Oropouche burden and land-use patterns in the different municipalities of the Rio de Janeiro state.

The results, detailed in the Results section, guide our discussion on the potential vectors facilitating OROV dissemination in Rio de Janeiro.

6.4) Add study limitations:

(i) sample restricted to the state of Rio de Janeiro and the 2024–2025 period;

(ii) underrepresentation of asymptomatic or undiagnosed cases;

(iii) reliance on official surveillance data, subject to underreporting;

(iv) possible bias in spatial representation (not all affected municipalities had genomes included).

Reply: The Discussion section was updated based on the reviewer’s suggestions.

6.5) Add future perspectives:

(i) importance of strengthening genomic surveillance to monitor potential reassortments, since OROV has a segmented genome;

(ii) need to integrate human, vector, and wild reservoir surveillance (One Health approach);

(iii) development of sensitive differential diagnostic tests for OROV, especially in regions endemic for dengue, ZIKV, and CHIKV, to avoid misdiagnoses;

(iv) putting the study into an international perspective: expansion beyond the Amazon reinforces that neglected arboviruses can become national and even regional challenges.

Reply: The Discussion section was updated based on the reviewer’s suggestions.

7. Page 17, line 366: Please correct the link to the epidemiological dashboard — there is an unintended space (gap) in the URL.

Reply: The text was changed accordingly.

8. Always use “Rio de Janeiro state” (or “the state of Rio de Janeiro”) consistently.

In some places, it simply appears as “Rio de Janeiro,” which can be confusing for the city.

Reply: The text was changed accordingly.

---

## [Reviewer Report · REVIEWERS COMMENTS]

## REVIEWER #1

The manuscript meets all the criteria requested for evaluation. Moreover, the authors have thoroughly addressed all reviewers’ comments and incorporated all necessary revisions.

The manuscript is now suitable for acceptance in MIOC.